# Determinants of Food Waste from Household Food Consumption: A Case Study from Field Survey in Germany

**DOI:** 10.3390/ijerph192114253

**Published:** 2022-10-31

**Authors:** Henrike Hermanussen, Jens-Peter Loy, Bekhzod Egamberdiev

**Affiliations:** 1Department of Agricultural Economics, University of Kiel, 24118 Kiel, Germany; 2Leibniz-Institute IAMO, 06120 Halle, Germany

**Keywords:** household food waste, avoidable food waste, food waste composition, determinants of food waste, Germany

## Abstract

According to FAO about one-third of the food worldwide is discarded. The economic, environmental, and social (ethical) impact of food loss and waste (FLW) is substantial. Food waste (FW) at the household level in high income countries makes a significant share of total FLW. Target 12.3 of the Sustainable Development Goals advocates a 50% reduction of the global per capita FW by 2030. The German government has agreed to this goal. Across all sectors, about half of the waste is avoidable. To achieve a reduction of FLW, information on the current level, its causes, and the economic costs of its reduction are necessary. Depending on the definitions and methodologies to measure FLW, studies have come to different results. This study estimates and analyses avoidable and total household FW and for the first time its determinants in Germany. On average, 59.6 kg per capita of food is wasted annually, of which 49% is avoidable FW. The main causes of household FW are eating habits, shopping behaviour, involvement in FW, and retail promotions.

## 1. Introduction

According to the Food and Agriculture Organization (FAO), about 1.3 billion tons of food, equivalent to one-third of worldwide food production, is lost or wasted. Along the value chain, food loss and waste (FLW) occur at every stage, including production, transportation, storage, processing, retailing, and consumption. Compared to developing countries, very few post-harvest losses occur in developed countries due to technical progress. However, in developing countries, about 40% of the total FLW comes from households [1]. Determinants are consumer preferences, lack of consumer knowledge, high income, socio-cultural norms, and certain other value judgments [2,3,4,5,6,7].

The United Nations (UN) FW Index Report indicates that on average 79 kg of food are wasted per capita in high income countries [8]. Politicians, society, and scientists alike increasingly recognise the problem of FLW. A great number of recent publications, many political initiatives and related international organisations’ statements underline the importance of FLW [3].

With the “Agenda 2030 for Sustainable Development” [9]., the international community has committed to effectively combat hunger and malnutrition (SDG 2) by reducing FLW [10,11]. The current crisis in Ukraine underlines the role of reducing FLW. Since “*The International Day Against Food Waste*” in September 2020 the public attention has been drawn to the agenda’s sub-goal 12.3, to “*halve per capita global food waste at the retail and consumer levels and to reduce food losses along production and supply chains, including post-harvest losses [by 2030]*” [10,12].

Negative environmental effects are caused by a high degree of eutrophication of water bodies, impairment of biodiversity, and increased CO2 emissions [1,5,6]. FLW reduction may as well be an appropriate measure to conserve resources [13]. Besides the environmental impact of FLW, considerable economic defects appear for society and individuals [1,4,5]. German consumers dispose of approximately 4.4 million tons of food every year, which equivalents a total value of 6 billion Euros. Correspondingly, a two-person household disposes of food worth about 150 Euros every year [12].

The reduction of FLW is the main objective of food policies in Germany [14,15]. However, empirical evidence on the extent, costs, and causes of FLW are rare and uncertain [7]. Moreover, uniform definitions and methodologies to analyse FLW are still missing [3,6,8]. For German households only limited evidence on FW is available. Thus, the aim of this research is to provide data on avoidable FW. We pursue two main objectives: (1) We estimate the amount and the value of avoidable FW in households. (2) We identify and measure the determinants of individual household FW by a multiple regression model. We utilise the most common methods to measure FW by carrying out a household questionnaire and diary study.

The paper is structured as follows: we start by stating objectives for reducing and definitions of FLW. We discuss the approaches to measuring FLW and propose a model to estimate the determinants of FW. In Section 3, we provide a brief overview of the literature. Section 4 presents the data collected for this study. In Section 5, the results of multiple regression models are presented. Finally, we summarise our findings, present some conclusions, and give an outlook to potential future research.

## 2. Objectives, Definitions, and Methodology

In the following, we discuss the objectives behind the reduction of FLW. This is necessary because measures and aggregation methods depend on objectives. In the Agenda 2030 [9], no objectives are presented and measures or aggregation are discussed.

### 2.1. Objectives for Reducing FLW

The following four objectives are most often cited in the literature [1,10,13,15,16,17]:(1)Food security is the main objective for many societies. According to the United Nations’ Committee on World Food Security, food security defines a situation in which all people, at all times, have physical, social, and economic access to sufficient, safe, and nutritious food that meets their food preferences and dietary needs for an active and healthy life. FLW reduction or prevention can potentially help to move up food security, in particular in developing countries where food security is often crucial. Nonetheless, donations to food banks can also contribute to the reduction of FLW and improve the food security of endangered groups.(2)Efficiency of resource use is a general and major objective for societies in a world of limited resources, in particular under strongly growing populations [1,6,17,18]. FLW appears to be a waste of resources. However, reducing FLW can incur costs; thus, zero FLW may be inefficient.(3)Environmental protection or economic sustainability is another important objective. Economic sustainability refers to practices for long-term economic growth without having negative impacts on social, environmental, or cultural levels. FLW often has negative effects on the environment, e.g., through landfills. The economic costs of landfills are mostly not incorporated in the cost of production or waste.(4)For many people, FLW also has a moral dimension. FLW is considered to be unethical. A lack of appreciation of discarded food is contemplated to be morally unacceptable by many people. For instance, wasted meat is disrespectful of the animal lives taken. Many cultures honour respectful and FLW-minimising handling of foods [1,19].

### 2.2. Definition of FLW

To define FLW, we need a definition of food. The regulation of the European Parliament (EC) No. 178/2002 stipulates the general principles and requirements of food law. In Article 2 it says: “*[…] food or foodstuff means any substance or product, whether processed, partially processed or unprocessed, intended to be, or reasonably expected to be ingested by humans*.” [20].

Based on this definition, FAO (1981) defines that “FLW is wholesome edible material intended for human consumption, arising at any point in the food supply chain that is instead discarded, lost, degraded or consumed by pets” [21]. More recently HLPE defines FLW as “a decrease, at all stages of the food (supply) chain from harvest to consumption, in mass, of food that was originally intended for human consumption, regardless of the cause” [6]. FLW can occur at the production, post-harvest, and processing stages of the food supply chain (FSC) [15] referring to a decrease of food for human consumption, regardless of the cause of their disposal [1,6]. These losses can be categorised according to their origin. A distinction is made between food loss (FL) and food waste (FW) [1,15,22] (see Figure 1). FW occurs at the end of the value chain at the level of household consumption, whereas FL occurs at previous stages of the supply chain [6,19,21].

FW can be further differentiated according to its potential of avoidance into: avoidable, facultative avoidable and unavoidable waste [15] (see Figure 2).

As in many industrialised countries, consumers in Germany are responsible for most of the FLW within the FSC. Consumers include households (small consumers) and large consumers such as canteens and restaurants. FW results from consumers’ behaviour and attitudes. Thereby it is also manipulated by retailing practices such as promotions and price and quantity discounts [19,24]. “Overeating” of consumers may also be considered as FW [25] (see Figure 3).

In this study we focus on the avoidable FW for edible parts that can be ingested by humans. We define FW as a decrease of the food intended for human consumption

### 2.3. Measuring and Aggregating FLW

Different estimation methods and metrics are used to measure FLW and aggregate FLW over the stages of the FSC [3]. The most common aggregation method uses quantities. Alternatively, FLW is measured by the values or calories. All measures result in different scales and shares (see Table 1).

Though FLW focuses on food, food items are always a combination of food and non-food components, such as transportation, packaging, labour, etc. The processing along the FSC changes the combination of these components. Wasting or losing a kg of potatoes at harvest is different from wasting a kg of potatoes at the household level. Though the costs of bringing the waste to the landfill may be similar, the loss of resources is higher for potatoes wasted at the household level. Values of FLW more adequately consider the use of resources at the various stages of the FSC [26].

### 2.4. Approaches to Estimate FLW


(1)In questionnaire-based surveys household members assess the amount of food wastage. Answers are often closer to social or individual norms than to true values [27,28,29]. According to the study by Schmidt et al. (2015), the amount of avoidable waste is up to 19% lower [2].(2)Waste composition analyses (WCA) focus on a sorting system for household waste. WCA are complex and expensive. Only FW disposed of via the municipal waste system is documented. The WCA have limits in identifying foods and their kinds [27]. The awareness of being part of an FW study may lead to behavioural changes (“*Hawthorne effect*”) [2,12].(3)In food waste diaries household members document the amount of FW during a certain period. FW diaries require the time and effort of participants. Results, however, may be comparatively more accurate than surveys. Households may still adjust their behaviour [30,31,32]. In contrast to WCA, FW diaries allow to analyse the waste composition and its determinants. Food fed to animals, disposed in the sewer, or composted is documented. Some household FW may even not be noted, e.g., food is taken to and wasted at the office or school [12]. Though often overrated, the share of unavoidable waste is estimated [13,33].(4)Meta-analyses are based on the results of other primary surveys or extrapolations. The main strength of this method is the access to large amounts of data, which could facilitate comparability.


Here, we combine questionnaire-based surveys and FW diaries to detect both reliable quantities and drivers of FW and to obtain information about households. The diary method is expected to be more accurate than the questionnaire. Participants of the diary-study document avoidable FW.

### 2.5. Causes of Consumer FW

Several studies show that household waste behaviour has various determinants and interdependencies [1,5,13,25,29]. Different driving forces like incorrect understanding of best-before-dates, little overview on stocks, poor time management, and insufficient planning which causes quality problems, such as little knowledge and competence of preparation as well as re-use of leftovers lead to FW. Moreover, children’s eating habits can also foster FW [5,6,13,34,35,36]. Further, consumer preferences, eating habits, and socio-economic settings can foster FW. Many consumers prefer freshness, variation, and variety of food. A lack of awareness and education increases FW [4,37]. In many high-income countries, food is relatively cheap and often additionally put on promotional sales by retailers. Due to specials, promotions and bigger pack sizes consumers feel forced to buy beyond actual demand [38]. Causes lead to the actual reasons why food is disposed of. Consumers state that they dispose of food due to qualitative deficiencies (e.g., freshness, brown spots on the apple, or dry bread), or due to exceeded best-before-dates. Food is often wasted because consumers fear foodborne diseases or consumers are unable to assess food born health risks. Consumers feel that wasting food even prior the best-before-date reduces health risks [13,25,29,34,39,40,41]. If, however, self-determination and freedom of choice are assumed for consumers, the observed FLW may be the result of utility maximising [41] (see Table 2).

### 2.6. A Benchmark Model of Household FW

To determine FW at the consumer level, we follow “*The Household Food Waste Journey Model*” by Principato (2018) based on a marketing and behavioural framework. By considering conscious and unconscious waste, the model justifies behaviour of consumers following a decision-making process of dealing with food [23].

Derived from this model we draft 5 variable clusters to determine the avoidable and the entire FW and formulate the following hypotheses (see Figure 4): (1) shopping behaviour: planned shopping, more frequent shopping, and closer distance to the supermarket reduce FW. As well as higher expenses for food and its quality lead to higher FW; (2) eating habits: the less frequently cooking is done in the household and the more ready-to-eat meals are consumed as well as more frequent restaurant visits are predictive for higher FW; (3) spoilage: the more likely consumers apply sensory examination of food, and the more consciously groceries are bought and stored the less food is wasted; (4) involvement: the greater the interest in the subject of FW and the more the disposal of food is considered morally reprehensible the less FW occur, whereas suffering from food poisoning leads to higher disposal rates; (5) grocery retail: retail sales promotions such as volume discounts or special price offers increase the FW.

Moreover, the influence of demographic conditions on the disposal of food is examined. To state our benchmark model, the consumer decision-making process used in marketing has been adapted for FW. Accordingly, it is assumed that psychological, social, situational, demographic, and socio-economic factors shape individual wasteful behaviour. These factors influence behaviour and every phase of the “*Household Food Waste Journey model*” [23]. This combines the various theoretical drivers for wasteful behaviour with their effects on the food management process (planning, shopping behaviour, storage/preparation, consumption, and disposal).

Due to available evidence in the literature, hypotheses are formulated in a targeted manner. The basis for the statistical hypotheses is a multivariable linear regression. This results in the following equation:y = β_1_ x_1_ + β_2_ x_2_ + … + β_j_ x_j_ + α + ε

Here, the dependent variable y describes the estimated amount of avoidable FW per week in grammes. The independent and various confounding variables are described by x_j_, while β_j_ stands for the regression weights. Moreover, α is for the additive constant and ε is for the random error.

## 3. Literature Review

Table 3 provides an overview of studies of FW. The estimations of FW are divided into total, avoidable, and estimated monetary losses. Most FW studies focus on Europe. Monier et al. (2010) present one of the first studies [48]. In 2014 Bräutigam et al. adopted the FAO’s approach to considering the same waste coefficients calculated for each European country with data from FAO Food Balance Sheets [35]. Recently, Stenmarck et al. (2016) [49] and Caldeira et al. (2019) [50] conducted a meta-analysis. The methodological criteria in the existing studies coincide and could be compared; therefore, the research field is fairly harmonised in Europe.

Many European countries conducted or compiled a single study to generate an FW baseline in order to reappraise the policy-set targets. In Germany, the baseline study with data from „*Gesellschaft für Konsumforschung*” was carried out by the Thünen Institute in 2015. Accordingly, a regular consumer panel to record FW in households was used in which the participants answered a questionnaire and participated in a diary study [12]. In a few countries, such as Norway [36], the Netherlands [51] or Slovenia [52], there are repeated data-gathering efforts for tracking FW over time. Only Belgium [43] and Poland [53] include studies of specific sub-national areas. Between countries, however, there is a substantial variation in methodology and assigned confidence level. A large number of studies measure only edible FW, using FW diary methodology.

Recent studies in the case of North American countries include aggregated national-level figures. To form a national average, the Canadian study [54] aggregates 56 WCA of household FW. The US estimations come from a recent Environmental Protection Agency paper [55], published to improve the methodology and calculated sector-specific surplus food production under waste management pathways.

As for measurement, different methods are applied. The recorded amounts of total FW, avoidable as well as unavoidable waste, reach from 39 kg [42] (Austria, method: WCA) up to 139 kg [42] (Europe, method: meta-analysis) by having the mean of annual 55 kg per capita. While avoidable FLW per capita and year range from 6.6 kg [27] (Italy, method: questionnaire-based survey) to 91.8 kg [56] (Switzerland, method: WCA), mean of 45 kg. On average, 45% of the total amount of waste is assigned to avoidable FLW. Studies rarely deal with other approaches besides quantity measurements. As a result, studies dealing with monetary losses are seldom. The annual avoidable monetary losses per person result in values ranging from 17.12 Euros [47] to 244.58 Euros [57].

A harmonisation of the research field is necessary in order to be able to compare results and derive political measures.

**Table 3 ijerph-19-14253-t003:** Literature overview.

Country	Study	Method^1^	Total FW /Capita and Year [kg]	Avoidable FW /Capita and Year [kg]	%	Avoidable FW/Capita and Year [€]
Austria	Environment Agency Austria, 2017 [42]	WCA	39	19	12	n.a.
Belgium	Flemish Food Supply Chain Platform for Food Loss, 2017 [43]	WCA	72.3	32.7	45	n.a.
Canada	Environment and Climate Change Canada, 2019 [54]	WCA	79	n.a.	n.a.	n.a.
Denmark	Danish Environmental Protection Agency, 2018 [44]	WCA	80.7	44	24.8	n.a.
	Edjabou et al., 2016 [45]	WCA	85	48	56.4	n.a.
	Tonini, D., Brogaard, L.K.-S., & Astrup, T.F., 2017 [58]	WCA	n.a.	46	n.a.	n.a.
Estonia	Moora, Evelin et al., 2015 [47]	D + Q	54	17	36	17.12
Europe	Caldeira et al., 2019 [50]	M + S	21–139	n.a.	n.a.	n.a.
	Monier et al. (2010) [48]	M + S	7–133	n.a.	n.a.	n.a.
	Stenmarck et al., 2016 [49]	M + S	91	71	60	n.a.
Finland	Silvennoinen et al., 2014 [59]	D	n.a.	23	n.a.	70
	Katajajuuri et al., 2014 [39]	D + Q	67	23	35.4	70
France	ADEME, 2016 [60]	S	85	24	28	n.a.
Germany	Cofresco, 2011 [61]	D + Q	80	47.2	59	182.9
	Jörissen et al., 2015 [27]	Q	n.a.	7.2 g	n.a.	n.a.
	Hafner et al., 2012 [5]	S	81.6	38.4	47	94–122.2
	Noleppa und Cartsburg, 2015 [19]	S	63	32.94	54	n.a.
	Schmidt et al., 2019 [12]	D + Q	54.5	27.0	44	75
Greece	Abeliotis et al., 2015 [62]	D	98.9	29.8	30,1	n.a.
Hungary	Kasza et al., 2020 [63]	D + Q	65.49	31.97	48.8	n.a.
Italy	Giordano et al., 2019 [33]	D + Q	67	27.6	41.2	n.a.
	Jörissen et al., 2015 [27]	Q	n.a.	6.6	n.a.	n.a.
Latvia	Tokareva, T., 2017 [64]	S	55	n.a.	22.7	237.78
Luxemburg	Luxembourg Environment Ministry, 2020 [46]	WCA	88.5	23.5	40.5	75.5
Netherlands	The Netherlands Nutrition Centre Foundation, 2019 [51]	WCA + Q	50	34.3	68.6	120
	Van Dooren, C., 2019 [65]	WCA + S + M	61.8	41.2	66.7	n.a.
Norway	Hanssen et al., 2016 [36]	WCA + Q	79	46.3	58.6	n.a.
	Williams et al., 2012 [38]	D + Q	n.a.	44.2	n.a.	n.a.
Poland	Steinhoff-Wrześniewska, 2015 [53]	WCA	50	n.a.	n.a.	n.a.
Portugal	Baptista, P., Campos, I., Pires, I., & Vaz, S., 2012 [28]	Q	31	n.a.	n.a.	n.a.
Romania	Dumitru et al., 2020 [29]	Q	n.a.	11.8	n.a.	n.a.
Slovenia	Vidic, T., & Žitnik, M., 2017 [52]	Q + S	73	27	36	n.a.
Spain	Ministerio de Agricultura, 2018 [66]	D + S	38.22	27	70	n.a.
Sweden	Swedish Environmental Protection Agency, 2014 [67]	WCA	81	28	35	n.a.
Switzerland	Beretta & Hellweg, 2019 [56]	WCA + S	n.a.	91.8	45	n.a.
United Kingdom	WRAP, 2020 [57]	WCA	100	68.5	68.5	244.58
USA	U.S. Environmental Protection Agency, 2020a [65]	M	59	n.a.	n.a.	n.a.

Remarks. FW = food waste; n.a.: not available. 1 method: D = Diaries; M = meta-analysis; Q = questionnaire; S = Statistics; WCA = waste composition analysis.

## 4. Data

The data collection uses two approaches: a questionnaire-based survey and an FW diary. Both approaches were pretested in a pilot study in 2017. The final questionnaire-based survey was conducted from August to September 2018 in two grocery shopping centers in the city of Kiel in Northern Germany. The two locations represent different income groups characterising a broad section of the population of Kiel. The market in the North of Kiel is frequented by wealthy and more affluent people. The other market in the Southwest of Kiel is mainly frequented by people from a low-income suburb. Participants were approached in the entrance area of the grocery shop. We estimate that 80% of the approached persons participated in the questionnaire-based survey. The questionnaire comprised 37 questions, focusing on shopping and eating habits, FW information, the management of groceries, and demographics. In total, nearly 668 questionnaires were completely answered.

Following the survey, we asked the interviewees to participate in a FW diary study. Participants received a voucher worth 20 Euros. In total, 62 participants (~10%) took part in the FW dairy study. In total, 53 respondents completed the study and could be matched up with the questionnaire-based survey. In total, 9 diaries could not be matched because of missing or false codes. In the FW diary, participants reported details on avoidable FW for two consecutive weeks. They also reported causes and channels of disposal. We also asked participants to collect groceries receipts to determine product prices.

According to Table 4, most of the participants (86%) live in Schleswig-Holstein. The gender distribution is 36% male and 64% female. In total, 81% of the participants are managing the household and are responsible for grocery shopping and planning of meals. On average, 2.4 persons live in a household (SD = 1.33); the average household size for Germany is 2.1 [68]. About 50% of the households have a net income between 500 and 2000 Euros. It is noticeable that households mostly spend 300–500 Euros on groceries despite higher income. In total, 80% (4%) of the participants consume meat (fish), 9% are vegetarian, and 2% are vegan.

Half of the respondents shop once or twice a week; 40% shop three to five times per week. The average distance between household and the nearest shopping center is 1.2 km. Neumeier (2014) finds that consumers in German urban agglomeration areas must walk 1.5 km to the nearest grocery shop [69]. Sample consumers prefer supermarkets and discounters, which corresponds to the market shares of retailers [70].

About 17% always use a shopping list, and 47% use it most of the time. Two thirds buy less food in advance to reduce their disposals. Spending as little time as possible on groceries is important to 38%; 42%, however, enjoy spending time shopping. The regional origin of products is relevant, 74% state that they are willing to pay a higher price for regional products, 82% are willing to pay higher prices for higher quality foods, and 35% try to save money on grocery purchases; 23% claim that food is expensive.

In total, 44% cook three to five times a week, and 40% cook every day. Most do not consume ready-to-eat meals. In total, 61% eat out of home once or twice a week, and 39% never eat out of home.

About 10% of the participants experienced severe food poisoning. FW is of great importance to the respondents: 89% are interested in the subject of FW, and more than 80% find FW morally reprehensible. All reported weights in the diaries in terms of single products are classified per category. For instance, “potato”, 20 g, is classified as a “vegetable”. Table 5 shows the statistics of model variables.

## 5. Results

Based on various statements, the participants provided individual causes of food disposal (see Table 6). Most of the participants (69%) stated that they still consume food after the expiration of the best-before date. Only 21% of them indicate that they dispose of food immediately after the best-before date. Most participants use appearance (77%), smell (81%), and taste (69%) to assess whether food should be disposed of. In total, 60% mention that the freshness of the food is not critical for disposal. Half of the respondents discard food when they have doubts about its safety. In addition, 20% of participants are often uncertain about the safety of food.

Table 7 shows further statements on the causes and driving forces behind household FW. Participants could mark multiple statements. Ten percent of the participants did not choose (agree to) any of the given state options. On average, participants mark two statements. Most often respondents state that they dispose of food because they could not process the food in time (38%). In total, 36% said that they cook too much. The third most frequent is that package sizes are too big (33%). In total, 32% note changing plans as a cause of wasting food.

The participants in the diary study most often note that they dispose of food due to spoilage (58%). In total, 15% dispose of food because the quality was not satisfactory. 12% (8%) agree that cooking too much (taste dislikes) cause food disposal. Another cause is an expired best-before date (5%). In total, 3% of participants name other reasons, e.g., food that falls on the floor or food that is burned.

The survey participants estimate the amount of FW per person and week. The amount of FW includes unavoidable and avoidable FW. Answers range from 0 g to 4 kg. The mean value per person and week is 1.145 kg (SD = 807 g). The proportion of unavoidable FW on average is 51% (SD = 27). This results in a total annual quantity of 59.6 kg per person, of which 29.2 kg are avoidable. Participants of the diary study estimate the total amount of FW in the questionnaires slightly higher at 62 kg, but report the share of avoidable FW at 35%. The avoidable FW recorded in the diary is aggregated over mass and monetary values. The values are calculated using cashier receipts and online prices (see Table 8).

As for the projections over the year, this corresponds to an avoidable amount of FW of approx. 29 kg per person. The determined amount corresponds almost exactly to the estimated avoidable amount of FW of the entire sample. The determined amount of FW estimated here is close to the German baseline study by Schmidt et al. (2019) [12]. In comparison to other studies for European countries these estimates appear to be rather low. However, the mass of FW is mostly underestimated due to socially desirable answers [15]. Often the disposal of food causes a bad conscience [30,71]. Our study also shows a discrepancy between the amount of FW reported in survey and diary studies. Participants in the diary study underestimate the FW by 24%. This might be caused by “socially desirable” behavioural patterns [2,37]. Furthermore, we may assume that the awareness of participants in a survey causes behavioural adaptations.

The aggregation of the monetary value shows that food worth 2.9 euros per week and person is disposed of. Extrapolated to one year, costs of 151 euros (per person) arise from discard. An estimation of the monetary losses was not carried out as part of the survey since test subjects in the pilot study had difficulties in answering them. The identified value is higher than those determined in other German studies [5,12]. The documented food expenditure per week is noted as 80 euros per week, 343 euros per month and 4165 euros per person and year. If the food expenditure is conjoint with food-related monetary losses, the average waste of food expenditure per person is 3.6%.

For given categories, including: vegetables, fruits, baked goods, meat, fish, dairy products, and others, it should be estimated how food is disposed (see Figure 5).

Ultra-fresh products and food without a best-before date like vegetables (17%), fruits (17%), bakery products (14%) and dairy products (14%) are most often discarded. In the diary, the proportion of fruit was ten percentage points higher and none of the diary keepers disposed fish. In addition, the “homemade meals” category has been added; 10% of FW is generated in this category. The diary also recorded whether food was disposed in a processed or unprocessed state. 23% of the amount of FW was disposed in a processed state, which was accompanied by a correspondingly higher input of energy and resources.

As for the diary, it is possible to determine the monetary losses incurred through disposal for each category. The results of the different aggregation methods are compared (see Figure 6). While the dispersion is roughly the same for most categories in terms of quantity and value, it is noticeable that there are major discrepancies for five categories: Although meat products make up a small part of the waste quantity, they account for a significantly larger share in terms of value. The same applies to the “homemade meals” and “miscellaneous” categories. The opposite is true for the vegetable and baked goods categories.

The following Table 9 depicts the multivariable regression results between the chosen predictors and the amounts of FW in quantities. Thereby, we estimate (as described in Section 2) 5 models each to extent the selected variables which determine the avoidable and the entire FW per person and week in kg at household level.

Our results match published findings in many instances. Variables of household and individual demographics were used in each model. Hereby it was found that the described relationship between demographic data and FW in the literature can largely be verified. The effect of gender and income have been contrarily discussed in the literature. This also holds for our studies, there are no significant results. The household size shows perennially a positive estimator—an increasing household size is predicted for an increase in FW, but the amount per capita is decreasing [12,27,38,61,72]. As described in the literature [4,12,15,29] age exhibits a negative prefix. It can be deduced that less food is thrown away by older people. Particularly people over 65 years waste less food. They may have a higher consciousness, less changing preferences, and a higher appreciation for food. Whereas the presence of children under 14 years causes higher FW [27,46,73]. Higher expenditures on food have a positive impact on food waste quantities. This might be explained by a correlation between household size and the tendency of buying too much. No link between a higher level of education and FW is found. This may be explained by increasing opportunity costs for better educated (high income) consumers [12,61,74,75].

As described in the model by Principato (2018) [23], different determinates ascertain the amounts of FW: variables of shopping behaviour such as a higher frequency of grocery shopping and a greater distance to the supermarket are causing higher FW. The incurrence of household’s opportunity costs corresponds to this consideration. Whereas consumers who pay more attention to paying little for their food dispose of less. Furthermore, for housekeeping skills such as the planning of purchases (shopping list) a relationship between increasing amounts of FW and a lack of planning results [1,27,61,65,74]. Those who are more frequently using a shopping list, reduce their amount of FW by 0.09 kg per week. Those who are trying to spend as little time as possible for grocery shopping show a reduced level of FW. This could be related to better planning of purchases or less variety of foods consumed. Those willing to pay higher prices for higher food quality are supposed to waste more food.

Eating behaviour such as a higher frequent consumption of ready-to-eat meals is expected to cause more FW and a higher frequency of cooking should lead to lower FW [12,61,74,75]. Our results indicate that people who cook more often produce more avoidable FW. This could be explained by the fact that consumers do not plan the quantities properly and cook too much and leftovers remain. Or the food has already been bought for cooking and plans change, e.g., by eating out with friends. The consumption of ready-to-eat meals increases the amount of FW. The more often the consumers are eating at a restaurant the more food is wasted. Thus, grocery shopping does not adequately account for the out of house consumption.

Following psychological factors, consumers often dispose of expired food without subjecting it to sensory testing [5,32]. However, only 5% of the diary study participants dispose of groceries because the best-before date is expired. In the survey, almost 70% stated that they consume food with an expired best-before date. However, the regression analysis demonstrates more spoilage occurs when consumers pay higher attention to the best-before-date. Further subjecting food to a sensory test, especially if consumers orientate themselves by the appearance or the taste of the food leads to higher AFW. Especially the orientation on the smell leads to less FW. According to the literature, food is often disposed of because of preferences for fresh food or out of concern for food safety. Consumers aim to reduce their risk of foodborne illnesses [30,71]. Further, consumers who consciously buy less food for stock waste less.

People with knowledge and involvement in the topic of FW show lower levels of food waste [38,74]. The topic of FW is of great interest to most participants (89%) but has shown no significant effect in the data under study. Besides, none of the variables of “involvement” show significant estimates. The greater the interest in the subject of FW and the more the disposal of food is considered morally reprehensible, the less FW is disposed of, whereas suffering from food poisoning leads to higher FW rates and lower proportions of avoidable FW.

Food retail sales-promoting measures, like special offers in terms of packaging sizes, as well as volume discounts and other marketing strategies, influence the amount of FW. Shopping behaviour is influenced in a targeted manner, which stimulates over-buying. Consumers buy these not needed offers to feel like they are getting more for their money [38]. Through clever positioning of products and packaging designs, retailers guide consumers to overbuy. Consumers often blame their overbuy to retailers’ marketing actions. Our study shows that the amount of FW decreases by independent portioning. If consumers can buy food in smaller portions, the amount of FW is reduced. It is questionable, however, if additional costs of packaging may level this effect. Figure 7 wraps up the study results. 

## 6. Conclusions

The role of consumers for FLW in economically developed countries is crucial. FW makes a significant share of FLW in these countries. To reduce FW, one needs to understand the reasons behind it. Our study, therefore, focused on estimating the quantities and values of avoidable FW in German households and identifying its underlying causes. The study produces many results similar to the literature, but also some differences. The survey study results an annual amount of (avoidable and unavoidable) FW of around 60 kg per person. The share of avoidable FW is about 49% or 29.2 kg per person per year. The German Government wants to reduce all FLW by 50%. This could be achieved if all avoidable FW is recovered.

The data of the diary study confirms an avoidable amount of annual FW of 28.2 kg per person. The participants in the dairy study estimate a lower amount of avoidable FW of 21.8 kg per person per year. The quantities are likely to be underestimated and may be considered as minimum values. According to the diary study, the annual monetary loss resulting from avoidable FW is estimated at 150 Euros per person per year.

Different causes are responsible for the disposal of household food. The main reasons for disposal are health risk reduction and utility maximisation. Disposal behaviour is shaped by social background, cultural influences, and socio-economic factors as well as phases of life. Waste is fueled by increased consumer choice and a shift in consumer preferences towards fresher products and higher quality, whereby “ultra-fresh” products and goods without a best-before-date dominate the FW. In addition, food retail practices, in particular food retail promotional activities foster household FW.

## 7. Policy Recommendations

In addition, significant determinants of avoidable FW were identified. The interest in the topic of FW has a positive influence on the amount of waste. Furthermore, the planning of grocery shopping reduces FW. The multiple regression analysis was able to determine mostly predictors of attitude variables and variables of shopping and eating habits. If further research confirms this result, measures against FW could consider the effects. It may be assumed that attitudes are easier to change than demographics.

To achieve a sustainable food system, policy measures can be divided into two categories: (1) information-based measures and (2) market-based and/or regulatory measures [76]. In addition, it is important to check whether there are market or policy failures in order to take appropriate corrective actions [26]. The theoretical genesis of household food waste behaviour should also be known to develop effective countermeasures [37].

(1)Information-based measures include government-sponsored public information campaigns aimed to raise public awareness of FW and promote sustainable consumption. This includes labels, certifications, and guidelines to educate consumers about the consequences of FW. This should enable sovereign and conscious consumer decisions and influence consumer preferences toward resource-efficient food consumption [3,5,77]. Changing consumer behaviour through direct communication about the need to reduce FW is crucial [6,78]. Usually, consumers do not react to appeals but are rather guided by incentives [26]. Personal benefits of reducing the waste rate, such as financial savings or “doing what is morally right”, should therefore be communicated. If consumers understand that wasting food is wasting money, FW may be reduced. Nonetheless, the costs of reducing FW should be considered as well. Furthermore, the health risks of consuming food after the best-before date or any bad food need to be taken into consideration. People need to be better prepared for such decisions. In the long term, school courses on the issue should be installed. In addition to such instruments, there are so-called “nudging” instruments. These try to indirectly influence and steer the consumer [79]. An example is a government-funded cooking class to improve housekeeping skills [23,80]. The conscious handling of food and food management is critical to reducing waste [6,38]. In this regard, the government should start some initiatives using social networks.(2)Market-based and regulatory measures include subsidies and taxes on specific foods or food components [80]. General sustainability goals in the food sector such as reducing land use and limiting greenhouse gas emissions must be clearly defined for regulatory measures [77]. A higher tax on meat, a food that takes a lot of resources to produce, would be an example. Furthermore, rewards may be paid to consumers reducing their waste by reducing their garbage pick-up fees. Product labels could also inform about the resource use in production and handling to make consumer more aware of the consequences of not using or wasting the product.

The previously dominant political measures in Europe are aimed at raising awareness and are voluntary, i.e., information- and education-oriented [81]. In context with the goals set in the “Agenda 2030” to halve global FLW, both the EU and the federal government have launched numerous campaigns aimed at drawing attention to FW. Thoughtfulness is paid to educating consumers; The consequences of FW are explained, and possible causes are discussed. For example, tips on avoiding waste in the household are given. A key component is information about the benefits of reducing food waste and valuing food for individuals and society. “Zu gut für die Tonne” [82] is a German federal government campaign in which digital media are used for communication and the topic of FLW is integrated into schools [78]. Despite such measures, the European Court of Auditors (2016) asserts that the EU is currently not effectively tackling the problem as discard rates remain high compared to EU baseline studies. Moreover, many potential improvements would not require new funds or initiatives, but rather a better adaption of existing policies would suffice. Obstacles related to food policy, food donation and sharing should be removed and legislation reinterpreted [80]. Fruit and vegetables that do not meet the specifications and standards, such as too curved cucumber, too pale tomatoes, or too small apples, are disposed of. These guidelines and regulations are often imposed by consumer preferences and their perfectionist demands. The European Council also insists on using food waste as animal feed. This point also is enshrined in the “Abfallrichtlinie der EU”. In addition, member states are asked in the “Kreislaufwirtschaftspaket” to: reduce FLW during production and distribution, in households, to encourage food donations and to repurpose surplus food. The action plan of the “Kreislaufwirtschaft” was proposed in 2015. It describes the EU’s plan to create a sustainable and competitive economy: the amount of waste should be minimised, and the value of resources should be preserved. FW is one of the five priorities identified in the action plan. The action plan summarises the environmental, social, and economic consequences of FW and the main countermeasures required. As part of this plan, FW quantities from landfills need to be reported to the EU annually [83].

The lack of uniform definitions and measurement methods makes it difficult to compare study results and setting targets [82]. The choice of survey methods has a strong influence on the results as well. A harmonisation of the research in this field is necessary to be able to compare different studies and data points. More research is needed on the costs and/or the potential loss in the utility of reducing food waste. In particular, health risks are difficult to evaluate. How many more risks may consumers accept if they reduce their FW by consuming outdated food products?

## Figures and Tables

**Figure 1 ijerph-19-14253-f001:**
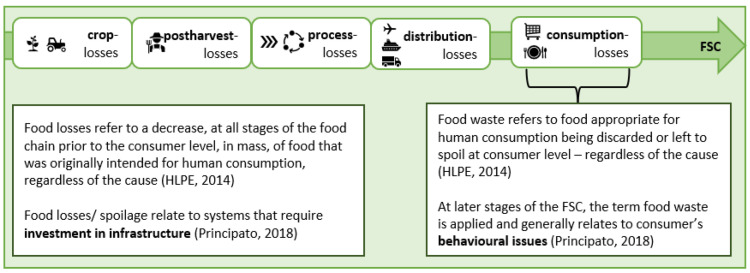
Food loss and waste definitions along the food supply chain (FSC) [6,23]. HLPE: High Level Panel of Experts on Food Security and Nutrition (HLPE) is the United Nations body for assessing the science related to world food security and nutrition.

**Figure 2 ijerph-19-14253-f002:**
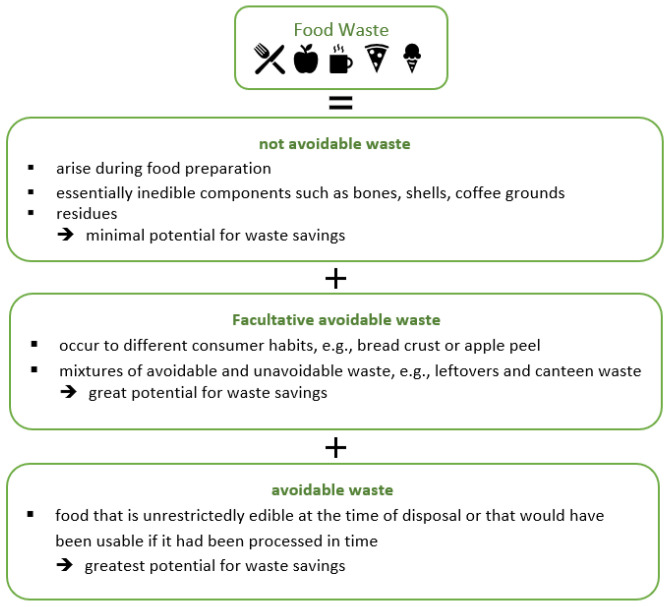
Food waste by avoidance potential. [5,7,12,23].

**Figure 3 ijerph-19-14253-f003:**
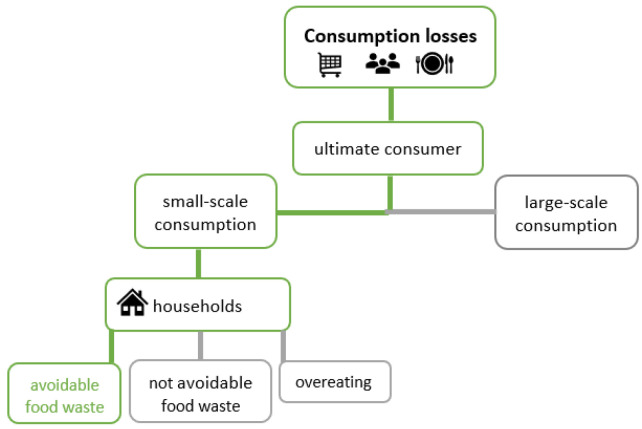
Consumption losses.

**Figure 4 ijerph-19-14253-f004:**
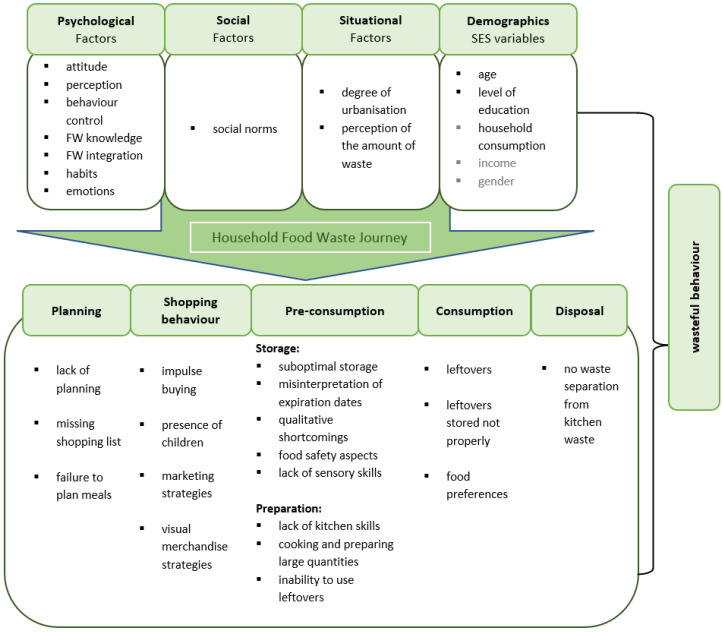
Model for explaining the generation of FW in households. Source: Own illustration. Based on Principato, 2018 [23]. Additions in grey letters.

**Figure 5 ijerph-19-14253-f005:**
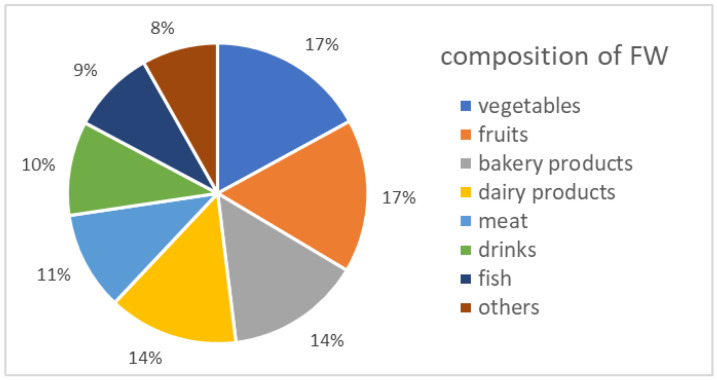
Questionnaire—estimated composition of food waste (aggregated in kg).

**Figure 6 ijerph-19-14253-f006:**
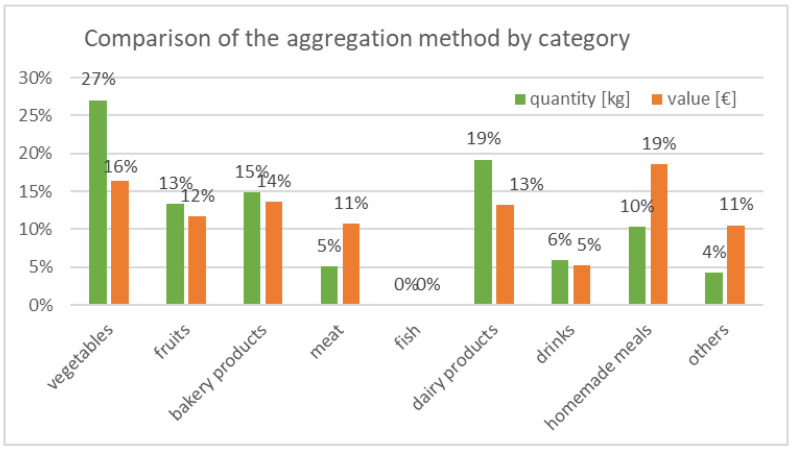
Diary—comparison of the aggregation method according to food category.

**Figure 7 ijerph-19-14253-f007:**
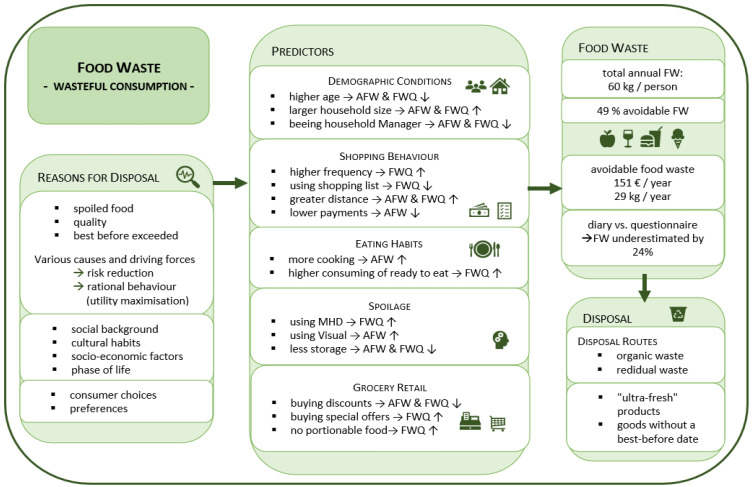
Influencing factors of food waste. Own illustration.

**Table 1 ijerph-19-14253-t001:** Comparison of different aggregation methods. [26].

Food Item	Mass [kg]	Value [€]	Calorie [kcal]
Potato	100	0.20	77
Bread	100	1.00	240
Butter	100	0.50	717

**Table 2 ijerph-19-14253-t002:** Reasons, causes, and driving forces behind disposal at consumer level.

Disposal Of Food At Consumer Level
Reasons for disposal ■ Best-before-date exceeded ■ spoilage: food is spoiled, possibly already at the time of purchase ■ qualitative deficiencies 🡪 risk minimisation: fear of foodborne diseases, health considerations, freshness 🡪 rational behaviour: maximising utility, minimising costs
Causes ■ food retailing - promotions - bigger pack sizes discounts (low price per gram) ■ consumers preferences: - freshness - taste (bought food does not taste good) ■ ethics, morals, culture, religion - eating habits (eat food with or without the skin) - lifestyle: “good provider” mentality (several different stocks in case of visitors) ■ socio-economic factors: income, educational level, family composition, … - e.g., children: picky eaters, food fights, awareness of freshness and high-quality food.
Driving forces of disposal—how does spoilage occur? ■ deficient/insufficient planning - quantity problems/demand uncertainty - poor time management (changing plans, weekly shopping) - no overview of stock ■ little knowledge/competence/skills - storage - preparation: portion size, cut away - re-use of leftovers - incorrect understanding of the best-before-dates - evidence of spoilage: organoleptic test

Source: Own compilation based on [1,4,5,6,13,19,20,29,42,43,44,45,46,47].

**Table 4 ijerph-19-14253-t004:** Descriptive statistics.

		Questionnaire N = 668	Diary N = 59
Age	Mean (SD)	43.1 (19.8)	40.3 (17.4)
Gender	male	36%	21%
	female	64%	79%
Household manager	yes	81%	94%
	no	19%	6%
Household size	M	2.4 persons	2.3 persons
	1 PHH	23%	21%
	2 PHH	44%	46%
	3 PHH	15%	21%
	4 PHH	12%	8%
	more than 5 PHH	6%	4%
	children > 14 years	15%	20%
	persons < 65 years	20%	14%
Level of education	secondary modern school (school year 5 to 9 in Germany)	6%	4%
	secondary school (school year 5 to 10 in Germany)	12%	2%
	final secondary-school examinations (qualifying for university)	24%	21%
	qualification/formation	22%	19%
	specialist	5%	0%
	bachelor	9%	25%
	master/state examination	21%	29%
Net household income	<500 €	9%	8%
	500–1000 €	21%	29%
	1000–2000 €	23%	16%
	2000–3000 €	19%	20%
	3000–4000 €	14%	10%
	>4000 €	13%	18%
Food spending	<100 €	3%	4%
	100–300 €	38%	37%
	300–500 €	41%	40%
	500–1000 €	17%	17%
	>1000 €	1%	2%
Diet	not vegetarian (meat-eating)	80%	67%
	pescetarian (no meat, but fish-eating)	4%	13%
	vegetarian	9%	12%
	vegan	2%	4%
	others	6%	4%

**Table 5 ijerph-19-14253-t005:** Descriptive statistics of the main variables.

Variables	Definition	Mean	S.D.	Min.	Max.
	**Dependent**				
FWQ	Food Waste per week/person (kg)	1.163578	0.8149499	0.05	5
AFW	Avoidable Food Waste per week/person (kg)	0.5983165	0.5825177	0.004	3.6
	**Shopping Behaviour**				
Frequency	Shopping frequency per week: 1 = less than 1×, 2 = 1–2×, 3 = 3–5×, 4 = daily	2.466877	0.6904956	1	4
Shoppinglist	Use of a shopping list: 1 = never, 2 = sometimes, 3 = frequently, 4 = usually, 5 = always	2.452681	1.134745	1	5
Distance	Distance (km) to the nearest grocery shop.	1.188368	1.855701	0.002	15
Time	I try to spend as little time as possible shopping: 1 (not agree at all)–7 (totally agree about)	3.933227	1.871337	1	7
Quality	For high quality food, I am willing to pay a higher price: 1 (not agree at all)–7 (totally agree about)	5.709524	1.564634	1	7
Payment	I take special care to pay little for groceries: 0 = rejection, 1 = approval	0.347619	0.4765927	0	1
	**Eating Habits**				
Cooking	Cooking in household every week: 0 = rejection, 1 = approval	0.8280757	0.3776125	0	1
Readytoeat	Consuming of ready-to-eat meals at home per week: 1 = 0×, 2 = 1–2×, 3 = 3–5×, 4 = daily	1.583596	0.6483292	1	4
Gorestaurant	Going to restaurant per week: 1 = 0×, 2 = 1–2×, 3 = 3–5×, 4 = daily	1.768139	0.7317192	1	4
	**Spoilage**				
UseMHD	After the best-before date has expired, I will no longer consume the food: 1 (not agree at all)–7 (totally agree about)	2.789889	1.896472	1	7
Visual	Orientation on the appearance of the food: 0 = rejection, 1 = approval	0.7661927	0.4235857	0	1
Smell	Orientation on the smell of the food: 0 = rejection, 1 = approval	0.8120063	0.3910161	0	1
Taste	Orientation on the taste of the food: 0 = rejection, 1 = approval	0.6840442	0.465263	0	1
Storage	I consciously buy less food in stock so that I have to throw away less: 0 = rejection, 1 = approval	0.6571429	0.4750414	0	1
	**Involvement**				
Interest	I am interested in the topic of food waste. 1 (not agree at all)–7 (totally agree about)	4.310726	0.7217968	1	5
Moral	It’s morally reprehensible to dispose of food. 1 (not agree at all)–7 (totally agree about)	5.749206	1.522682	1	7
Poison	I already had severe food poisoning. 0 = rejection, 1 = approval	0.2333333	0.4232887	0	1
	**Grocery retail**				
Discount	I would prefer if stores reduce individual products prices instead of offering volume discounts. 1 (not agree at all)–7 (totally agree about)	4.283677	0.8225221	1	5
Specialoffer	I take advantage of special offers thus I buy more than I consume. 0 = rejection, 1 = approval	0.1822504	0.3769448	0	1
Singlebuy	I prefer to buy fruit, vegetables, and meat individually thus I can determine the amount by myself. 0 = rejection, 1 = approval	0.8288431	0.3769448	0	1
Portionable	If I cannot buy fruit, vegetables, or meat individually, I often buy more than necessary. 0 = rejection, 1 = approval	0.3708399	0.4834129	0	1
	**Control Variables**				
Age	Age of propositus (years)	43.3	19.74585	11	87
Female	1 if propositus is female, 0 otherwise	0.6396825	0.480474	0	1
Hhhead	1 if propositus is household manager, 0 otherwise	0.8079365	0.3942357	0	1
Diet	Nutrition 1 if Omnivore, 0 otherwise	0.7968254	0.4026811	0	1
Education	propositus level of education 1–7 the higher, the higher the level of education	4.205742	1.86461	1	7
Householdsize	Household size, persons per Household	2.401587	1.335226	1	13
Income	Total net income of the household 1 = >500 €, 2 = 500–1000 €, 3 = 1000–2000 €, 4 = 2000–3000 €, 5 = 3000–4000 €, 6 = <4000 €	3.485	1.530909	1	6
Children	Number of children; age ≤ 14	0.2587302	0.668181	0	4
Elderly	Number of seniors; age ≥ 65	0.2936508	0.6363082	0	3
Expenditure	Total grocery spending’s of the household: 1 = >100 €, 2 = 100–300 €, 3 = 300–500 €, 4 = 500–1000 €, 5 = <1000 €	2.745192	0.8156629	1	5

**Table 6 ijerph-19-14253-t006:** Assessments of causes of food disposal of food using a Likert scale from 1 to 5 (1–2: I do not agree to 4–5: I agree).

Statement	Mean	SD	I Do Not Agree (%)	I Neither Agree or Disagree (%)	I Agree (%)
After the best-before date has expired, I do not consume food.	2.8	1.9	69	11	21
I decide to waste food based on its appearance.	5.51	1.7	13	10	77
I decide to waste food based on its smell.	5.7	1.67	11	7	81
I decide to waste food based on its taste.	5.15	1.84	18	13	69
For me, enjoyment comes first, so I dispose unsightly food.	3.15	1.95	60	14	27
If I have the slightest doubt about the quality and safety of food, I dispose it immediately.	4.37	2.18	38	12	50
I am often not sure whether the food is still safe to consume.	3.08	1.79	62	15	23

Legend: SD = standard deviation.

**Table 7 ijerph-19-14253-t007:** Causes and driving forces behind food disposal (multiple answers).

Statement	%
• I do not have time to process food.	38%
• I cook too much; the leftovers are disposed.	36%
• The package sizes are too big, so I buy more than I need.	33%
• I go grocery shopping (for a week) and then my plans change.	32%
• I am often uncertain if food is still ok for consumption.	26%
• The groceries bought are already spoiled.	13%
• I do not like the taste of the food.	11%
• I buy too much because of special offers.	11%
• I do not check my groceries before I go shopping.	8%
• I shop groceries once a week and buy more than I actually need.	6%
• I buy discounted items which best-before date is almost up.	3%

**Table 8 ijerph-19-14253-t008:** Food waste estimates by survey and dairy study.

Amount of FW per Person	Survey Study [kg] All Participants	Survey Study [kg] Diary Participants	Diary Study [kg]	Diary Study [€]
FW per year	59.6 kg	62.1	-	-
Unavoidable FW per year	30.3 kg (51%)	40.3 (65%)	-	-
Avoidable FW per year	29.2 kg (49%)	21.8 (35%)	28.64 kg	151.0 €

**Table 9 ijerph-19-14253-t009:** Regression results.

		AFW			FWQ		
		Coef.	Std. Err.	*p* > |t|	Coef.	Std. Err.	*p* > |t|
Shopping Behaviour	Frequency	0.0567435	0.0442351	0.200	0.1101797 **	0.0502352	0.029
	Shopping list	−0.0437219	0.0324863	0.179	−0.0959333 **	0.0352713	0.007
	Distance	0.0348813 **	0.0146014	0.017	0.0536469 **	0.0193817	0.006
	Time	−0.0153615	0.01539	0.319	−0.0063845	0.0183956	0.729
	Quality	0.0215182	0.016665	0.197	−0.0015171	0.0207716	0.942
	Payment	−0.1137951 **	0.0569415	0.046	−0.0663974	0.0681687	0.330
	R^2^	0.0701			0.1193		
Eating Habits	Cooking	0.1295911 *	0.0662317	0.051	0.01574	0.1050921	0.881
	Ready to eat	0.0410228	0.0425748	0.336	0.215158 **	0.0543633	0.000
	Go restaurant	0.0108869	0.0414721	0.793	0.0164217	0.0535087	0.759
	R^2^	0.0500			0.1055		
Spoilage	UseMHD	0.0110118	0.0152312	0.470	0.0352006 **	0.0179512	0.050
	Visual	0.1667701 **	0.0620412	0.007	0.054637	0.0971448	0.574
	Smell	−0.0112435	0.0821608	0.891	−0.0053929	0.1191314	0.964
	Taste	0.0796165	0.0596496	0.182	0.0664339	0.0756688	0.380
	Storage	−0.1477875 *	0.0770927	0.056	−0.1915847 **	0.0723721	0.008
	R^2^	0.0671			0.0981		
Involvement	Interest	0.0063523	0.0334971	0.850	−0.0480371	0.0477593	0.315
	Moral	−0.013789	0.0183408	0.452	−0.0214964	0.0242163	0.375
	Poison	−0.0154726	0.0135059	0.252	0.0082132	0.0202242	0.685
	R^2^	0.0472			0.0832		
Grocery Retail	Discount	−0.0642994 **	0.0313529	0.041	−0.0989234 **	0.0394768	0.012
	Special offer	0.0033995	0.070435	0.962	0.1675357 *	0.0919937	0.069
	Single buy	−0.0644083	0.1149349	0.575	0.140901	0.0870997	0.106
	Portionable	−0.0106574	0.0579162	0.854	0.1229374 *	0.0685739	0.074
	R^2^	0.0529			0.1004		

Remarks. M = mean; SD = standard deviation; AFW = avoidable Food Waste per person/week (kg); FWQ = Food Waste Quantity per person/week (kg); ** *p* ≤ 0,05; * *p* ≤ 0,1 (*N* = 668). Source: Own.

## Data Availability

The data can be obtained from the authors upon request.

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
