# Peer review of "Determinants of Food Waste from Household Food Consumption: A Case Study from Field Survey in Germany"

_ijerph, 2022, doi:10.3390/ijerph192114253_

Round 1
Reviewer 1 Report
The study presents an interesting but well-covered field in literature; the quantification and determinants of food waste at German households using two methodologies. In this line, it is important to support the novelty that this article provides in this field. Therefore, the manuscript must be reviewed to meet the level and the standards of the Journal. The topic will be of interest to the readers of the journal.
Major comments
Line 17 Please include information about the novelty of your research.
Line 61 Please include the objectives of your research.
Please re-structure your manuscript. Materials and methods include definitions and also a literature review. Please be more concise in the description of the literature review.
Please define if you will show results and discussion in separate sections or in a same section of results and discussion.
I recommend the following order: Introduction (context and purpose of the study), literature review (main antecedents that support the study), material and methods (how the study was performed and statistical tests used), results (the main findings), discussion and conclusions (brief summary and potential implications). In this line, the section of results should only show results and conclusions should be re-written according to the objectives of the study.
Minor comments
Line 32 You provide six references of determinants. Please include the questions that emerge from the analysis of this literature.
Lines 45-47 I did not understand the association of the two sentences. Please be more precise about how FLW can be and adequate measure for the resource conservation. For example, biodiversity.
Lines 50-51 Please include the questions that emerge from the analysis of this literature.
Lines 52-53 Please analyze in more detail this information that maybe important to justify your research.
Author Response
Report 1
The study presents an interesting but well-covered field in literature; the quantification and determinants of food waste at German households using two methodologies. In this line, it is important to support the novelty that this article provides in this field. Therefore, the manuscript must be reviewed to meet the level and the standards of the Journal. The topic will be of interest to the readers of the journal.
Major comments
Line 17 Please include information about the novelty of your research.
- Thank you for pointing this out to us: our research builds on a previous study by the Schmidt et al. (2019). However, for the first time we use regression analysis to test determinants of avoidable and total FW in German households. We added this information to the abstract and to the introduction.
Line 61 Please include the objectives of your research. Please re-structure your manuscript. Materials and methods include definitions and also a literature review.
- Thank you for this important advice, which we follow closely by restructuring the paper. Also, considering comments of other reviewers, we decided to use the following outline. Even though, you recommend to start with the literature review, we start with the objectives and definitions, as these are needed to comprehend some passages in the lit review.
Please be more concise in the description of the literature review. Please define if you will show results and discussion in separate sections or in a same section of results and discussion.
- We reviewed these sections accordingly.
I recommend the following order: Introduction (context and purpose of the study), literature review (main antecedents that support the study), material and methods (how the study was performed and statistical tests used), results (the main findings), discussion and conclusions (brief summary and potential implications). In this line, the section of results should only show results and conclusions should be re-written according to the objectives of the study.
- We followed your advice.
Minor comments
Line 32 You provide six references of determinants. Please include the questions that emerge from the analysis of this literature
- We clarify those questions for our survey; this would take up too much space in the introduction.
Lines 45-47 I did not understand the association of the two sentences. Please be more precise about how FLW can be and adequate measure for the resource conservation. For example, biodiversity.
- We reformulate the sentence: „ Negative environmental effects are caused by a high degree of eutrophication of water bodies, impairment of biodiversity, and increased CO2 emissions [1, 5, 6]. FLW reduction may be an appropriate measure to conserve resources and reduce external effects [12]. “
Lines 50-51 Please include the questions that emerge from the analysis of this literature.
- We want to evaluate the German household FW and its determinants. The literature review allows us to put our results in order.
Lines 52-53 Please analyze in more detail this information that maybe important to justify your research.
- Thank you for pointing this out to us: We revised the passage as following: „ The reduction of FLW is a main objective of food policies in Germany [14, 15]. However, empirical evidence on the extent, costs, and causes of FLW are rare and un-certain [7]. Moreover, uniform definitions and methodologies to analyse FLW are still missing [3, 6, 8]. For German households only limited evidence on FW is available. Thus, the aim of this research is to provide data on avoidable FW. We pursue two main objectives: (1) We estimate the amount and the value of avoidable FW in households. (2) We identify and measure the determinants of individual household FW by a multiple regression model.”

Reviewer 2 Report
Determinants of Food waste from household food consumption: a case study from field survey in Germany
Comments
Introduction
Please cite the exact publication of FAO that observed 1.3 billion tons of food being lost. Check the first sentence under Introduction.
Methodology/Data
Data quality control and standardization, representativeness checks should also be provided to validate the sample.
Authors should provide the implicit/explicit model of the regression analysis carried out. All the variables should be defined with their respective units of measurement.
Policy Implications and Recommendations
Please add “Policy Implications and Recommendations” as a section after Conclusions. Policy implications refer to what the results and findings and conclusions mean for policies and programmes of public sector and non-state actors. Policy implications are suggestive and indicative of broad and specific directions and instruments that can be used to address observed gaps, utilize observed opportunities and correct existing anomalies or inadequacies. Recommendations are what should be done, who should do it and in what conditions and for what. Recommendations are action-based and should be in the form of actionable measures. Recommendations should be actionable, sharp, unambiguous, reflective and specific.
As a guide, endeavour to construct a set of policy implications around each conclusion or set of conclusions. With a good set of conclusions and policy implications, the stage is set for the identification of actionable recommendations.
Provide policy recommendations with actionable items for government, research institutions, development partners, multilateral/bilateral organizations, urban planners, private sector initiatives and civil society interventions. Remember, no recommendations should be made that does not derive from the policy implications.

Author Response
We addressed all comments. For easier reading we kept the original comments in our reply
Report 2
Introduction
Please cite the exact publication of FAO that observed 1.3 billion tons of food being lost. Check the first sentence under Introduction.
- We did cite the source at the end of the pargraph. To make this more clear, we also cite the source after the first sentence. [1] Gustavsson, J.; Cederberg, C.; Sonesson, U. Global Food Losses and Food Waste: Extent, Causes and Prevention ; Study Conducted for the International Congress Save Food! At Interpack 2011, [16 - 17 May], Düsseldorf, Germany; Food and Agriculture Organization of the United Nations: Rome, 2011; ISBN 978-92-5-107205-9.
Methodology/Data
Data quality control and standardization, representativeness checks should also be provided to validate the sample. Authors should provide the implicit/explicit model of the regression analysis carried out. All the variables should be defined with their respective units of measurement.
- The sample is a random draw of customers in the field. As the initial survey has a response rate of about 80 percent, we feel very confident about the random nature. The household size fits to the size in the general population.
Policy Implications and Recommendations
Please add “Policy Implications and Recommendations” as a section after Conclusions. Policy implications refer to what the results and findings and conclusions mean for policies and
programmes of public sector and non-state actors. Policy implications are suggestive and indicative of broad and specific directions and instruments that can be used to address observed gaps, utilize observed opportunities and correct existing anomalies or inadequacies. Recommendations are what should be done, who should do it and in what conditions and for what. Recommendations are action-based and should be in the form of actionable measures. Recommendations should be actionable, sharp, unambiguous, reflective and specific. As a guide, endeavour to construct a set of policy implications around each conclusion or set of conclusions. With a good set of conclusions and policy implications, the stage is set for the identification of actionable recommendations.
Provide policy recommendations with actionable items for government, research institutions, development partners, multilateral/bilateral organizations, urban planners, private sector
initiatives and civil society interventions. Remember, no recommendations should be made that does not derive from the policy implications.
- We have restructured the paper and included now a section conclusions and policy recommendation where we take up this point in detail.

Reviewer 3 Report
1) There have been a lot of previous empirical studies on the factors affecting the amount of recyclables (domestic waste and food waste). These previous studies should have been systematically reviewed and used to construct the conceptual framework.
2) Is there any incentive-related factor in the model in Figure 6? Given the collective-action nature of food waste recycling, economic incentive is very important to motivate people to recycle. The authors should search relevant literature using the keywords "collection action", "recycling" and "economic incentive".
3) What sampling strategy were used in the survey? Is the sample representative enough?
4) Are the research findings generalizable to other parts of Germany (or even other European cities)?
5) What are the limitations of the research? How do these limitations affect the interpretation of the research findings?
6) What are the contributions of the paper in the development of theories?
7) The paper has quite many typos and grammatical errors. The authors should have the paper proofread by a professional English writer before submission.
8) Minor issue of plagiarism was spotted. Particular attention should be paid to sources #3 and #5 in the attached iThenticate report.

Author Response
We addressed all comments. For easier reading we kept the original comments in our reply
Report 3
1) There have been a lot of previous empirical studies on the factors affecting the amount of recyclables (domestic waste and food waste). These previous studies should have been systematically reviewed and used to construct the conceptual framework.
- We have provided a comprehensive literature review with a focus on developed countries in chapter 3 now. Based literature review, we build a Food Waste Journal Model by Principato (2018) and drive hypotheses from it.
2) Is there any incentive-related factor in the model in Figure 6? Given the collective-action nature of food waste recycling, economic incentive is very important to motivate people to recycle. The authors should search relevant literature using the keywords "collection action", "recycling" and "economic
incentive".
- The model in Figure 6 does not include incentive-related factors. It demonstrates factors and behaviours that cause food waste at the consumption level. From this, incentive-based measures could be derived. We have included some thoughts on this in the policy recommendations section.
3) What sampling strategy were used in the survey? Is the sample representative enough?
- The survey was run two food retail outlets. Customers were approached randomly. 80 % of the approached agreed to participate. Therefore, we can assume a random same. We also did some comparison with population statistics which confirms that the sample represents the population. If necessary, we can provide more details on this.
4) Are the research findings generalizable to other parts of Germany (or even other European cities)?
- Overall, the sample characteristics match those of the German population and can therefore be regarded as representative of the German population. The literature review shows to what extent German results compares to other developed countries.
5) What are the limitations of the research? How do these limitations affect the interpretation of the research findings?
- All surveys have the main limitations that they may show intended but not actual behaviour. To test this, we also did a dairy study, which makes consumers to directly measure their own food waste instead of recalling it. However, the second approach may have the disadvantage that people for the time of observation may change their behaviour towards their intentions. We discuss these quite general problems in the paper.
6) What are the contributions of the paper in the development of theories?
- We develop an empirical model to explain household FW based on the literature. We also discuss the costs and potential utility losses of FW, which have not clearly been addressed in the literature.
7) The paper has quite many typos and grammatical errors. The authors should have the paper proofread by a professional English writer before submission.
- We intensively proof read the paper.
8) Minor issue of plagiarism was spotted. Particular attention should be paid to sources #3 and #5 in the attached iThenticate report.
- We have checked the respective passages, which referred to papers that have been coauthored by the authors of this paper. We have rephrased some passages.

Round 2
Reviewer 3 Report
The authors have addressed the reviewers' comments to my satisfaction. I don't have any further comment on the paper.